# Redox Potential and Crystal Chemistry of Hexanuclear Cluster Compounds

**DOI:** 10.3390/molecules26113069

**Published:** 2021-05-21

**Authors:** Elena Levi, Doron Aurbach, Carlo Gatti

**Affiliations:** 1Department of Chemistry, Bar-Ilan University, Ramat-Gan 5290002, Israel; Doron.Aurbach@biu.ac.il; 2CNR-SCITEC Istituto di Scienze e Tecnologie Chimiche “Giulio Natta”, Sezione di via Golgi, via Golgi 19, I-20133 Milano, Italy; 3Istituto Lombardo Accademia di Scienze e Lettere, via Brera 28, I-20121 Milano, Italy

**Keywords:** redox potential, Lever’s parameters, metal–metal bond, bond valence sum, steric effect, effective ionic charge

## Abstract

Most of TM_6_-cluster compounds (TM = transition metal) are soluble in polar solvents, in which the cluster units commonly remain intact, preserving the same atomic arrangement as in solids. Consequently, the redox potential is often used to characterize structural and electronic features of respective solids. Although a high lability and variety of ligands allow for tuning of redox potential and of the related spectroscopic properties in wide ranges, the mechanism of this tuning is still unclear. Crystal chemistry approach was applied for the first time to clarify this mechanism. It was shown that there are two factors affecting redox potential of a given metal couple: Lever’s electrochemical parameters of the ligands and the effective ionic charge of TM, which in cluster compounds differs effectively from the formal value due to the bond strains around TM atoms. Calculations of the effective ionic charge of TMs were performed in the framework of bond valence model, which relates the valence of a bond to its length by simple Pauling relationship. It was also shown that due to the bond strains the charge depends mainly on the atomic size of the inner ligands.

## 1. Introduction

One of the main questions often arising in the studies of redox potential of metal–organic complexes is whether their electrochemical behavior can be predicted through the properties of the metal and the isolated ligands [1]. This question is particularly pertinent for the TM_6_-cluster compounds (TM = transition metal) (Figure 1). It was shown that a high lability and diversity of the outer (apical or terminal) ligands in these compounds allow for tuning of redox potential and the related spectroscopic properties in wide ranges [2,3]. The choice of the inner (capping or bridging) ligands is also crucial for the potential values. However, a mechanism of the potential tuning is still unclear. To elucidate this mechanism, in this paper we will discuss possible correlation between structural and electrochemical properties of cluster compounds.

A possibility of such correlation is based on the fact that a large part of cluster compounds is soluble in polar solvents, in which the cluster units maintain their atomic structural arrangement. As a result, the electrochemical experiments, in particular cyclic voltammetry, became one of the ways to characterize structural and electronic features of solids. As an example, Figure 2 presents the redox potential of [Fe_6_S_8_(PEt_3_)_6_]^*n*+^ (*n* = 0–4) in dichloromethane solution as a function of the Fe-ion charge or formal oxidation state. Goddard et al. showed that each species from *n* = 0 to *n* = 4 can be a part of new solid phase with its own crystal and electron structures, e.g., [Fe_6_S_8_(PEt_3_)_6_], [Fe_6_S_8_(PEt_3_)_6_] (BPh4) etc. [4]. Thus, for cluster compounds, the redox transition in solution is intimately related to the respective changes in solids. The upper inset in Figure 1 shows the cyclic voltammetry (current vs. voltage) obtained for this system, with four oxidation and four reduction peaks, which take place upon electrochemical process. The redox potential, *E*_1/2_, is commonly calculated as a mean value between oxidation and reduction peaks for each transition between two species with different valence states. The black points on the main graph correspond to the formal oxidation states of the Fe atom upon transition between two redox-states, e.g., the average between Fe oxidation states in [Fe_6_S_8_(PEt_3_)_6_]^4+^ and [Fe_6_S_8_(PEt_3_)_6_]^3+^. In contrast, the red points are related to the initial species, e.g., to [Fe_6_S_8_(PEt_3_)_6_]^4+^ (see the reaction in the Figure 2). The average Fe valence is more appropriate to characterize the redox transition, but the use of the valence for one of the species introduces only a systematic error, which is commonly not important for comparison of the redox potentials in different compounds. For example, for the Re_6_-, Mo_6_- and W_6_-compounds with 24-electron cluster, it is usual to assign the redox potentials to the initial compound that losses one of the electrons in the electrochemical processes (24/23e redox transition) [5], in spite of the fact, that the potential is related to the mixture of two species.

Figure 2 illustrates a general relationship: the higher the transition metal (TM) charge, the higher is the potential of the redox transition and the more difficult it is to oxidize the cluster compound. Thus, it is clear that the redox potential of a cluster compound depends on the TM oxidation state, but it is affected also by the ligand composition [2,3,7]. Pombeiro summarized the efforts to establish the ligand effect on the redox potential for coordination compounds [8]. In his review Pombeiro recommended to use a simple correlation proposed by Lever [9]:*E*_1/2_ = *S_M_* [Σ *E_l_*] + *I_M_*(1)
where *S_M_* and *I_M_* are tabulated constants for the redox couple of a given metal, which depend on the spin state and stereochemistry, but, in organic solvent, are insensitive to the net charge of the species. *E_L_* are the electrochemical parameters tabulated for about 200 different ligands [9].

Szczepura et al., used the Lever’s approach to explain the change in the redox potential in three cluster compounds [Re_6_Se_8_(PEt_3_)_5_L]+ (L = I; 5-methyltetrazolate, 1,5-CH_3_CN_4_; and acetonitrile CH_3_CN) [10]. Plotting the potential values of the complexes (measured vs. ferrocene couple FeCp_2_^+^/FeCp_2_) versus the *E_L_* sum for the six terminal ligands, they obtained the linear correlation for the 24/23e redox transition:*E*_1/2_ [Re(III)_5_Re(IV)/Re(III)_6_] = 0.378 Σ *E**_l_* + 0.282     (V)(2)
where the numbers are not related to the Re metal, but rather to the cluster core, [Re_6_Se_8_]^2+^. Similar approach was used by Yoshimura et al., who showed that for complexes, [Re_6_S_8_Cl_4_L_2_]^2−^ (L = N–Heterocyclic ligands), the redox potential depends linearly on the *pKa* values (the analogs of *E_L_*), decreasing for more basic ligands (with higher *pKa*) [2]. This correlation was explained by the effect of *σ* electrons of the ligands on the HOMO level of the Re_6_-cluster. According to the authors, the potential growth for the ligand series L = dmap < mpy < py < bpy < cpy < pz (dmap = 4-dimethylaminopyridine; mpy = 4-methylpyridine; py = pyridine; bpy = 4,4-bipyridine; cpy = 4-cyanopyridine; pz = pyrazine) suggests that the coordination of electron-withdrawing groups make the [Re_6_S_8_]^2+^ core more difficult to oxidize. Sasaki marked that the increase of *E*_1/2_ related to substitution of Cl− ligands by pyridine is associated with reduction of electron density on the cluster core due to the less effective *π*-donation ability of pyridine as compared to that of Cl− [11].

To determine the first redox potential, *E*_1/2_ [Re(III)_5_Re(IV)/Re(III)_6_], in a series of cluster compounds, [Re_6_L^i^_8_L^a^_6_]^4^− (L^i^ = S or Se inner ligands, and L^a^—various negatively charged terminal ligands with different donor-acceptor character), Rojas-Poblete et al. performed calculations based on the Born–Haber thermodynamic cycle and using relativistic DFT for estimating free-energy values. [12]. According to the authors, the redox potential generally correlates well with the bonding energy calculated previously for the same ligands [13]. The higher the absolute value of the negative energy, the lower is the potential: F− < I− < Cl− < Br− < NCO− < NCS− < NC− < CN− < OCN− and more stable should be the cluster complex, while the redox potentials for Cl−, Br− and I− are relatively close to each other.

In 2001 Gabriel et al., summarized nicely the available values of the redox potentials for Re_6_-, Mo_6_- and W_6_-cluster compounds [5]. However, they marked the difficulty to derive out any general trend upon the ligand effect on these values. Indeed, many questions related to the redox potential still remain open. Is Equation (1) proposed by Lever really relevant for estimating the ligand effect on the redox potential in the TM_6_-cluster compounds? Do the inner and outer ligands affect differently the redox potential? If yes, why does it happen? How does the presence of different ligands affect the electron density distribution inside the cluster? Literature screening shows that none of the approaches mentioned above used structural data to explain the redox properties of cluster compounds. Moreover, as far as we know, none of the works took into account the effective ionic charges of TMs, which differ considerably from the formal ones in the case of compounds with metal–metal bonds [14]. Recently, using crystal chemistry approach, we described the effect of the ligand surrounding on the effective ionic charges of TM in the solid cluster compounds [15,16]. However, additional study is needed to investigate whether the results of these works are also relevant for the redox process in solution.

Thus, in this paper we analyze the available electrochemical data, with the aim of separating the different effects, which may affect the redox potential. Then, we calculate the effective charges of the TMs in solids and correlate them with the electrochemical data of the same cluster units in solution. To explain the obtained results, we present a short description of the crystal chemistry of the TM_6_-cluster compounds. For calculation of the ionic charges, we used the Bond Valence Model (BVM) [17], which relates the interatomic distances to the bond valences by simple Pauling equation [18]. For ionic solids, coordination compounds etc., it is well known that the BVM provides accurate values of the ionic charges in the form of bond valence sum (BVS) [19]. In our previous works we showed that the BVM is also very effective in the case of cluster compounds [20,21].

## 2. Methods: Calculations of the Effective Ionic Charges (or BVS) of TMs in Cluster Compounds

The effective ionic charge of the TM in a given cluster compound can be calculated as BVS of the metal–ligand bonds based on the lengths of these bonds, *R _TM-L_*, and respective empirical constants, *R_0 TM-L_* [22]. To avoid any problem in the choice of these parameters, we used an alternative way, namely, the method based on the bond order or valence conservation in these type of cluster compounds (See Section 3.3.1) [23]. Hence, the calculations were performed by the following formula:*V_TM_* = *N _TM_* − Σ *s _TM-TM_*(3)
where *N _TM_* is the number of valence electrons of TM (7 for the Re atoms and 6 for the Mo and W atoms. Note that in compounds under study all the electrons are bonding). Σ *s _TM-TM_* is the BVS of the TM-TM bonds where each term of the sum is given by:*s _TM-TM_* = exp[(*R_0 TM-TM_* − *R _TM-TM_*)/*b _TM-TM_*].(4)

Here, *s _TM-TM_* and *R _TM-TM_* are the valence and length of the bond, respectively. *R_0 TM-TM_* and *b _TM-TM_* are the bond valence parameters, transferable for the TM-TM pair in different compounds. These parameters were established in our previous works based on quantum chemistry and crystal chemistry data: *R_0_ Re-Re* = 2.495 Å, *b Re-Re* = 0.26 Å; *R_0_ Mo-Mo* = 2.51 Å, *b Mo-Mo* = 0.34 Å; *R_0_*
*W-W* = 2.535 Å, *b W-W* = 0.29 Å [23].

The advantage of this method as compared to the original BVM mentioned above is the universality of the bond valence parameters for all the Mo_6_-, W_6_- or Re6-cluster compounds, which increases the accuracy of the BVS comparison for different complexes. In addition, the crystal structures of most of cluster compounds are known from the X-ray experiments, in which the position of the heavy atoms is determined much more accurately than that of the relatively light ones. Thus, it is preferable to use the TM-TM distances in the BVS calculations.

## 3. Results and Discussion

### 3.1. Analysis of Available Electrochemical Data

#### 3.1.1. Case of the Constant Cluster Core and Different Outer Ligands

The first question that we would like to clarify is the validity of the Lever’s parameters for cluster compounds. Correlation between the redox potential and the sum of the Lever’s parameters for different series of the Re_6_-cluster compounds is presented in Figure 3. The values of the redox potentials are taken from refs. [5,10,24,25]. The electrochemical experiments were performed in CH_3_CN or CH_2_Cl_2_ solutions. The first two series can be described by the formula Re_6_S_8_Cl_6_-xLx, in which the outer Cl ligands are partly replaced by N–Heterocyclic (py, ppy = 4-phenylpyridine, bpy, mpy, bpe = 1,2-bis(4-pyridyl)ethan, pz) or triethylphosphine (PEt3) ligand (L) groups. The larger the number of these groups, x, the higher is the redox potential. In the third series, Re_6_Se_8_Cl_6_-xLx, the sulfur in the cluster core is replaced by Se, while the outer ligands are represented by various organic and inorganic atoms, groups and their combinations: I−, CN−, CH_3_CN, CO, PEt_3_, 1,5-MeN_4_C and dppm = 1,1-Bis(diphenylphosphino)methane. In the last series, the Re_6_Te_8_ -core is surrounded by CN− or CNCH_3_ ligands.

As can be seen from Figure 3 for each series, the redox potential is linearly related to the sum of the Lever’s parameters. Thus, Equation (1) is really valid for each series, but it is not universal for all of them. At first glance, it seems natural that the redox potential is affected by the kind of the inner ligands. It is logical to suggest that S, Se and Te have different values of the electrochemical parameters *E_L_* (which we did not find in literature), and this difference in *E_L_* is responsible for the separate correlations, *E*_1/2_*(E_L_)*, for the complexes with the Re_6_S_8_-, Re_6_Se_8_- and Re_6_Te_8_-cores in Figure 3. However, this suggestion contradicts with other electrochemical data. For example, the redox potentials of Mo_6_S_8_(PEt_3_) and Mo_6_Se_8_(PEt_3_) measured for three redox transitions (19/20, 20/21 and 21/22e) in different solutions (THF and CH_2_Cl_2_) are almost identical, and in some cases *E*_1/2_ of the selenide is even a little bit higher than that for the sulfide core [5,26]. Another strange feature is the separate correlations, *E*_1/2_*(E_L_)*, for the first two series, in spite of the same Re_6_S_8_-core. These obvious deviations of the universal Equation 1 strongly suggests that there is an additional factor besides the Lever’s parameters of ligands that should be taken into account to rationalize the redox process. In order to clarify this, we have to continue the analysis of the *E*_1/2_ data.

#### 3.1.2. Case of the Fixed Lever’s Parameter for the Outer Ligands

In this section we would like to show how different position of the same ligand atom affects the redox potential. For this, we chose Mo_6_-, W_6_- and Re_6_-halides and chalcogen-halides. All three halogen ligands have close values of *E_L_*: −0.24 for Cl−, −0.22 for Br− and −0.24 for I− [9]. Thus, we can expect that the redox potential of these compounds for the same TM will be independent of *E_L_*. In fact, *E*_1/2_ for compounds with the same cluster core [TM_6_L^i^_8_] and different outer ligands are very close (Figure 4. The electrochemical data are taken from ref. [5]). However, variation in the type of the inner ligands results in drastic potential changes. For example, substitution of one of the inner Cl ligands in Mo_6_Cl_14_ (not shown in Figure 4) by S or Se leads to the potential drop from 1.36 to 0.56 V. In this case, the drop might be caused by the difference between the Lever’s parameters *E_L_* for Cl− and chalcogenide ligands, but the replacement of Cl− by I− for the W_6_-cluster decreases *E*_1/2_ from 0.93–0.99 to 0.56–0.57 V. This potential drop is certainly not related to the Lever’s parameters, because the *E_L_* of Cl− and I− are exactly the same. Thus, it should be another reason for the variations in the redox potential, but from the electrochemical data it is not clear why the change in the position of the halogen ligands from the outer to the inner sites is so crucial for the redox process. Hence, the next step of our study was the analysis of ionic charges in cluster compounds.

### 3.2. Correlation Between the Effective Ionic Charge (BVS) of TMs and the Redox Potential

In this section, our aim was to reveal the effect of ionic charges (taken as BVS) of TMs on the redox potential, by excluding that of the Lever’s parameters. For this, we used the electrochemical data, similar to those of Section 3.1.2. We correlated them with ionic charges calculated for solids with the same cluster units as in solutions (Figure 5). For example, to describe the effective ionic charge of the Mo atom in the [Mo_6_Cl_8_Cl_6_]^2−^-cluster units, we calculated the respective BVS in three cluster compounds, (Bu_4_N)_2_[Mo_6_I_8_I_6_], Cu_2_[Mo_6_I_8_I_6_] and Cs_2_[Mo_6_I_8_I_6_]: 2.404; 2.487 and 2.438 v.u. These values show that, in spite of different cations in the three compounds, the BVSs of the Mo atoms are relatively close to each other and differ effectively from the formal value of 2. The results of calculations based on the TM-TM distances for all cluster compounds are presented in Appendix A.

A first feature that strikes the eye in Figure 5a is a huge difference in the potential of two series of the Re_6_-complexes. This difference can be explained by very different *E_L_* values of the outer ligands: 0.02 for cyanide CN− and 0.37 for methyl isocyanide CH_3_NC. In addition, all the data of Figure 5 show obvious correlation between the redox potential and the effective ionic charge calculated for the respective cluster compounds. This correlation is the most evident from Figure 5a. For each of two series of the Re_6_-complexes, the effective ionic charge increases with sulfur substitution by Se and Te. The higher the Re effective charge, the lower is the redox potential. (Note the TM BVS charge has nothing to do with the usual notion of atomic charge in quantum chemistry: rather it is related to the TM-L bond covalency, i.e., the larger the BVS charge, the greater the TM-L bond covalency and, as a consequence, the lower the redox potential.) Similar potential drop with ionic charge can be seen for the Mo_6_- and W_6_-cluster complexes (Figure 5b). Some point dispersion for these compounds is caused by the variety of the electrochemical data presented by different authors. For instance, according to Gabriel et al., *E*_1/2_ vs. SCE in CH_3_CN is equal to 1.36 V for [Mo_6_Cl_8_Cl_6_]^2^− [5], while Ebihara et al., and Jackson et al., presented the value of 1.56 [30] and 1.46 V [31], respectively. Nevertheless, it is clear that for the inner ligands the effective ionic charge increases from Cl via Br to I, with respective drop of the redox potential, while for the outer ligands this effect is much less pronounced. For example, for the compounds, (Bu_4_N)_2_Mo_6_Cl_8_L^a^_6_, the ionic charge of the Mo atoms increases from 2.953 for L^a^ = Cl to 2.962 for L^a^ = Br and to 3.067 for L^a^ = I. Thus, the redox potential of cluster units is clearly affected by the effective ionic charge of TMs in respective solids. Rationalization of these charges and the reason of their deviation from the formal values is presented in the next section.

### 3.3. Steric and Electrostatic Effects in Cluster Compounds and Their Influence on the Redox Potential

#### 3.3.1. Basic Crystal Chemistry of Cluster Compounds

To explain the results presented above we have firstly to remind the unusual crystal chemistry of the cluster compounds. In general, lattice strains caused by steric mismatch between different atoms in their packing arrangement is a typical phenomenon for a large part of solids. The bond strains result in the deviation of the BVSs from the formal values [32], but in compounds without metal–metal bonds these deviations are commonly very small. Additional metal–metal bonds change drastically this picture. The length of these bonds, corresponding to the formal bond order, is in most cases too short to match other bonds in the atomic packing. For the compounds under study a virtual naked octahedral cluster with a single TM-TM bond (24-electron cluster) is in most cases much smaller than a void formed by closely packed ligands. The larger the anion, the bigger is the void, available for this cluster, and the higher is the cluster/void mismatch.

Due to this mismatch, in order to realize bonding, the metal–metal bonds should be stretched, while the metal–ligand bonds should be compressed. For the TM_6_-clusters, the main factor that predicts the stretching of the TM-TM bonds (and the cluster size) is the size of the inner ligands, while influence of the outer ligands on the cluster size is relatively small [33]. The bond strains result in the valence discrepancy. It was shown that due to the high strains, the difference between effective and formal bond orders may be very high (about 1 v.u.) [20,34]. Nevertheless, due to redistribution of electron density around TMs, the valence (or bond order) deficiency in the metal–metal interactions is commonly compensated for by valence excess in the metal–ligand bonds (bond order conservation principle) [23]. As a result, the total BVS of TMs is equal or very close to the formal number of their valence electron. In general, in compounds without metal–metal bonds, the lattice strains impact the material instability [17]. In contrast, the bond strains in compounds under study are associated with more symmetric distribution of the electron density around TMs, stabilizing cluster units [23].

Another structural peculiarity of the TM_6_-cluster compounds is a high concentration of positive ionic charge in the same void. This concentration results in a very special distribution of the negative charges around clusters [35]. To illustrate this, the inset of Figure 6 shows the BVSs of the I atoms in (Bu_4_N)_2_Mo_6_I_8_I_6_. In this type of compounds, the inner and outer ligands are bonded to three and one TM atoms, respectively (See Figure 1). Since the lengths of the TM-L bonds in both cases are very close, the BVS of the inner ligands is about three times higher than that of the outer ones. Figure 6 shows also the anion BVSs or, more precisely, the contribution of the Mo_6_-cluster to the anion BVS for a number of cluster compounds with Mo_6_L_8_-core (L = Cl, Br, I). The BVS are presented as a function of the anion distance from the cluster center. According to the BVS drop, the TM_6_-cluster can be regarded as a virtual cation with high positive charge [16], while the inner ligands efficiently shield the outer ones from this charge. Interestingly, the BVS distribution is almost independent of the type of the outer ligands. For example, for (TBA)_2_Mo_6_Br_8_(CF_3_COO)_6_, the BVS of the inner (Br) and outer (O) ligand atoms is 1.91 and 0.57 v.u., respectively.

#### 3.3.2. Rationalization of the Factors Affecting the Redox Potential of Cluster Compounds

Now let’s highlight a few points that should be explained based on the above knowledge of crystal chemistry of cluster compounds:According to Equation (1) proposed by Lever, in organic solvents, the constants, *S_M_* and *I_M_*, are insensitive to the net charge of the species [9]. Why is this not the case for the compounds under study? In other words, why Equation (1) in our case is not universal for the redox couple of a given metal? In which cases will the redox potential depend solely on the Lever’s parameters?Why does the position of a given ligand in the cluster surrounding (inner or outer site) change effectively the redox potential of cluster compounds?Which ligand property is responsible of the changes in the effective ionic charge of TM?

The last two questions are the simplest ones. As was mentioned in the previous section, the bigger are the anions in the cluster surrounding, the larger is the void formed by these anions and the higher the mismatch between the unstressed cluster and the void. Consequently, the higher is the difference between formal and effective ionic charges (BVS) of TMs. Finally, it is the size of the ligand atoms that predicts the effective ionic charge and its deviation from the formal value. Thus, the difference in the *E*_1/2_ (BVS) correlations that we saw for various series in Figure 5 is caused by different radii of halogen or chalcogen atoms: 1.67 Å for Cl−, 1.82 Å for Br−, 2.06 Å for I−, 1.70 Å for S^2−^, 1.84 Å for Se^2−^ and 2.07 Å for Te^2−^ [36]. Due to geometric and electrostatic shielding, it is the size of the inner ligands that is the most effective in this case, while the effect of the outer ligands is rather minor. Nevertheless, the distinction of the redox potential for two series of [Re_6_S_8_Cl_6_-xLx]^*n*−^ that we saw in Figure 3 could be assigned to the difference in the size of the N and P atoms in the respective outer ligands, which is evident from the Re-L distances: 2.16–2.22 Å for L = N and 2.46–2.49 Å for L = P.

In order to answer the first question, we have to consider that Equation 1 was proposed for compounds without metal–metal bonds. As was mentioned above, for such compounds the BVS of TMs are commonly close to the formal values. In this case we can really expect that the constants, *S_M_* and *I_M_*, will be insensitive to the net charge of the TMs species, because this charge is almost constant. They will be different only for the different redox couples of a given metal (see Figure 2). In the case of cluster compounds the effective ionic charge of TM differs effectively from the formal one. It means that the net charge for a given couple is not constant, but changes with the ligand size. In this case Equation (1) proposed by Lever will be valid only for equal or at least for a close size of the inner and outer ligands in all the cluster compounds in the series.

## 4. Conclusions

In this paper, we analyzed the electrochemical data available in literature, in order to clarify the factors, which affect the redox potential of cluster compounds in polar solutions. Based on the previous crystal chemistry works and calculations performed in this study for respective solids, it was shown that the effective ionic charges (taken as BVSs) of TMs in cluster compounds differ effectively from the formal values. Due to the bond strains, the BVS of TMs are mainly governed by the size of the inner ligands. Thus, it was proved for the first time that the potential for a redox couple of TM depends not only on the TM constants and the electrochemical parameters of the ligands, as proposed by Lever for compounds without metal–metal bonds, but also on the effective ionic charge of the TM in a given cluster compound. Knowledge of the structure/TM charge relationship in cluster units should allow for more effective tuning of the redox potential and of the respective spectroscopic properties.

## Figures and Tables

**Figure 1 molecules-26-03069-f001:**
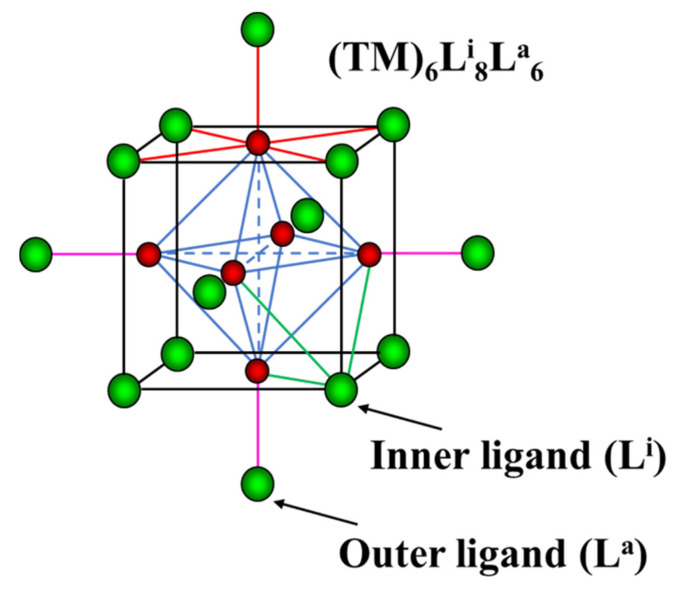
Cluster unit TM_6_L^i^_8_L^a^_6_. The TM and the ligands are in red and green, respectively. TM-TM bonds in octahedral clusters are marked in blue. The TM-L bonds around TM, inner and outer ligands are in red, green and pink, respectively.

**Figure 2 molecules-26-03069-f002:**
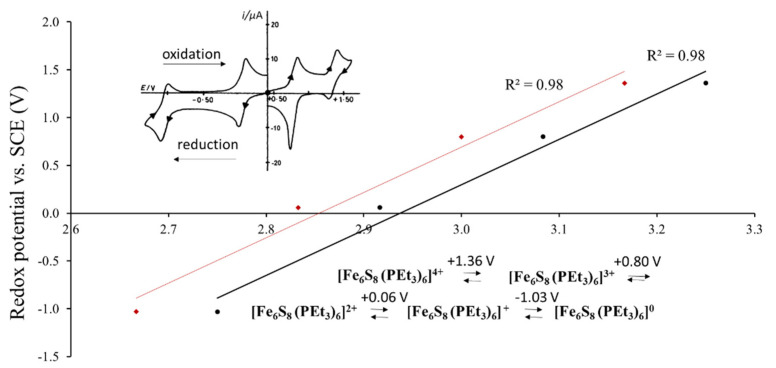
Redox potential of [Fe_6_S_8_(PEt_3_)_6_]^*n*+^ (*n* = 0–4) (vs. saturated calomel electrode (SCE)) in dichloromethane solution as a function of the formal charge (oxidation state) of the Fe-ion in the points of the redox transition (in black) and in the initial species (in red) (According to the electrochemical data of refs. [4,6]). The upper inset shows the cyclic voltammetry (current vs. voltage) with four oxidation and four reduction peaks associated with redox transitions upon electrochemical process. The low inset presents the chemical formula of the cluster species with respective potential, *E*_1/2_, which is the mean value between oxidation and reduction potentials for each redox transition.

**Figure 3 molecules-26-03069-f003:**
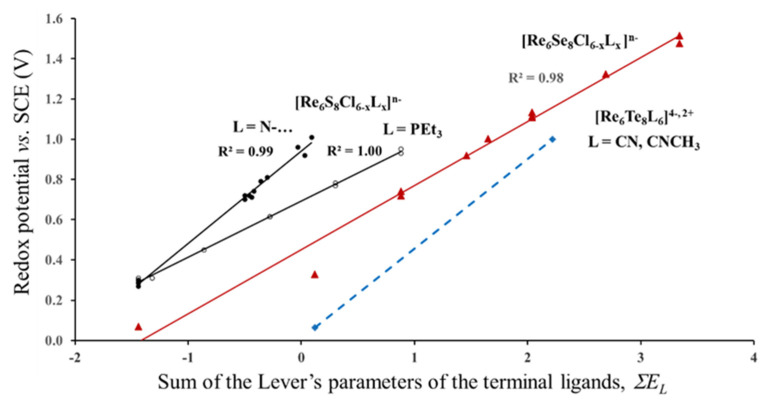
Redox potential (23/24 e−) of different series of Re_6_-complexes as a function of the Lever’s parameters of terminal ligands. The data for the Re_6_S_8_-, Re_6_Se_8_- and Re_6_Te_8_- cores are marked in black, red and blue, respectively. (According to electrochemical data of refs. [5,10,24,25]).

**Figure 4 molecules-26-03069-f004:**
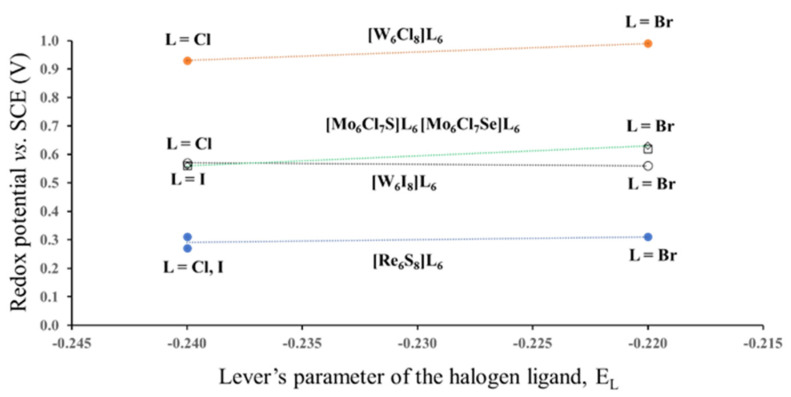
Redox potential (23/24 e−) of some Mo_6_-, W_6_- and Re_6_-halides and chalcogen-halides as a function of the Lever’s parameters for Cl, Br and I terminal ligands. The lines are the guide to the eyes. (According to the electrochemical data of ref. [5]).

**Figure 5 molecules-26-03069-f005:**
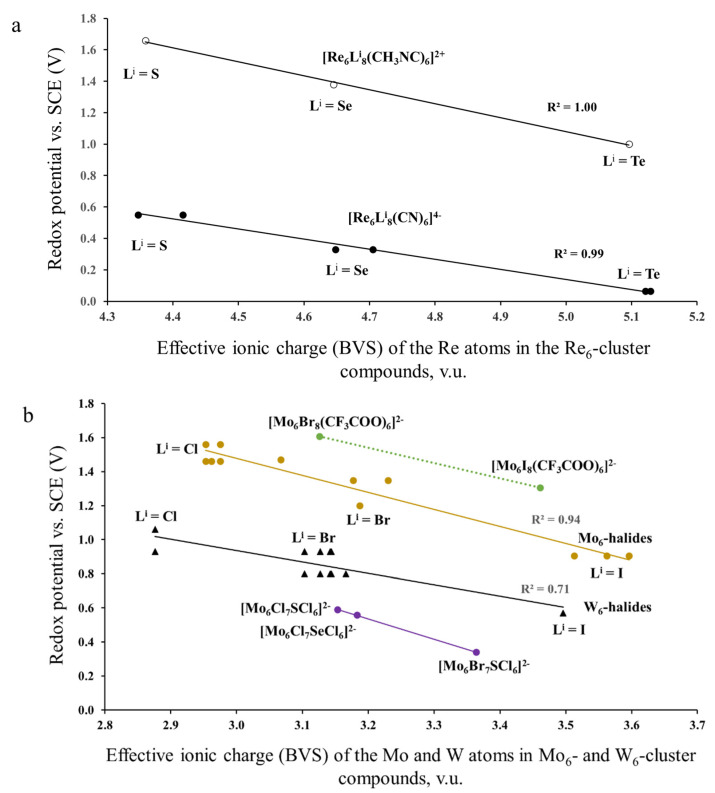
Redox potential (23/24 e−) as a function of ionic charges (BVS) of TM in Re_6_—(**a**) and Mo_6_—, W_6_—(**b**) cluster compounds. The electrochemical data are taken from refs. [27,28] for the Re_6_-complexes, ref. [29] for the Mo_6_-complexes with CF_3_COO ligands and refs. [5,30,31] for other compounds.

**Figure 6 molecules-26-03069-f006:**
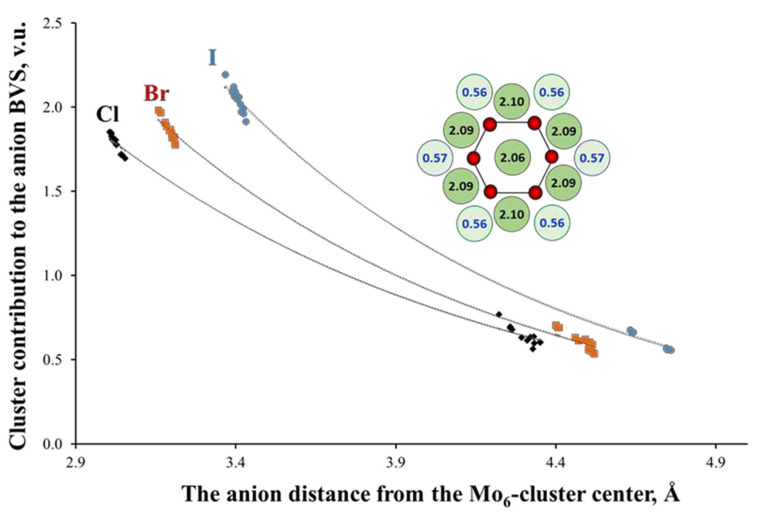
Mo_6_-cluster contribution to the ligand valence as a function of their distance from the cluster center in cluster compounds with Mo_6_L_8_-core (L = Cl, Br, I). The inset shows the arrangement of the Mo (in red) and I (in green) atoms in (Bu_4_N)_2_Mo_6_I_8_I_6_ (projection of the Mo_6_I_8_I_6_ cluster unit along one of the three-folded axes of the octahedral Mo_6_-cluster). The numbers are the BVSs of the I atoms around Mo_6_-cluster with high and low values for the inner (in dark) and outer (in pale) ligands, respectively (One of the inner ligands is located behind the central I atom).

## Data Availability

Not applicable.

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
