# Peer review of "Redox Potential and Crystal Chemistry of Hexanuclear Cluster Compounds"

_molecules, 2021, doi:10.3390/molecules26113069_

Round 1

Reviewer 1 Report

This article is devoted to the analysis of the electrochemical data available from literature. The article is well-written and can be interested for the specialists in electrochemistry and chemistry of cluster compounds. In my opinion, this article can be published in the Molecules without corrections.

Author Response

We greatly appreciate the positive judgement of the Referee on our paper. 

Reviewer 2 Report

This manuscript presents the relationship between the redox potentials and component ligands in hexanuclear cluster anions based on the literature data. I think that the analysis may be still a little rough, but it gets to the point. Overall works seem to be performed well, and therefore, this paper can be accepted for publication subject to the authors addressing the following comments.

(1) The authors should draw the molecular structure of “TM6-cluster” as shown in Figure 2 in Introduction.

(2) 5-9th lines on page 2 “The red and black points ...”: I don’t understand the meaning of this sentence.

(3) Inset of Figure 1: The authors should show the units in both axes, though they explain the units in main text.

(4) 8-10th lines from the bottom on page 2 “As can be seen from ...”: This would be too general a conclusion from this figure.

(5) “tetrazolate” on page 3 should be changed to “5-methyltetrazolate”.

(6) Equation 2: The authors should show the unit (i.e., volt) and the reference electrode (i.e., SCE). In relation, a full nomenclature of SCE must be placed just at the point of the first mention.

(7) Pages 3 and 4: Full nomenclatures of mpy, py, bpy, cpy, pz, ppy, and bpe must be described.

(8) 4th line from the bottom on page 3: “Recently” seems to be inadequate, because refs. 13 and 14 were published as early as 2007 and 2013, respectively.

(9) The authors should unify the notation of acetonitrile, either MeCN or CH3CN. If they describe it with the intention of methyl isocyanide, it should be CH3NC.

(10) Figure 3: The authors should show the unit in the ordinate axis.

(11) Figure 4b: “Mo6Br8(CF3COO” and “Mo6I8(CF3COO” should be changed to “Mo6Br8(CF3COO)6” and “Mo6I8(CF3COO)6”, respectively.

(12) 3rd-5th lines on page 8 “Some point dispersion for these compounds ...”: The authors should be more specific about the differences.

(13) Inset of Figure 5: I don’t understand the schematic structure of Mo6I8I6 (“(Bu4N)2Mo6I8I6” in the caption should be “Mo6I8I6”). Why does the Mo6 cluster have a hexagonal structure instead of an octahedral structure and why is the number of outer iodide indicated as a pale blue circle 13 instead of 14?

Reviewer 3 Report

This paper report interesting findings in the redox behaviour of transition metal cluster compounds and looks to understand changes in redox potentials for the clusters where ligands are changed throughout keeping a cluster constant in each case. The major outcome was the redox couple of these clusters not only is affected by transition metal constants and parameters of the ligands but additionally on the effective ionic charge of the metal in a particular cluster. This will be of interest to those studying electrochemistry and spectroscopy of clusters, but also provides knowledge towards the understanding of materials like these.

I feel this paper provides sufficient interest to those studying this area, and should be suitable for publication in its present state.

Author Response

(The authors gave the same response as above.)

Reviewer 4 Report

Electron transfer ability and redox chemistry of molecular forms of electron-rich cluster compounds are at the heart of a fruitful initiative that aim at properly re-assemble these superatoms into metal-organic constructs, especially for chalcogenide species taken off extended solids prior to exploring their fate once made available in polar solvents.  This post-high-temperature solid state chemistry initiative continue to be rewarded by a rich superatomic solid state chemistry  and physics. Yet, just how to tune the redox character thereby the amount of charge transferred remain a recurrent core issue throughout this journey. This is precisely the purpose of this paper. It is timely as the author note, supertonic solids are becoming influential. The paper will attract attention in the field.

Main remark: the authors stress that the effective charge localized on the TM6 core if key to control the redox potential.  It seems to me there is an opportunity for real significance further advances by exploring how  the electronic parameters provided by the g-factor of the 23 e- species are affected by the effective ionic charge, and its distribution over the 8 inner sites. For an investigation of the anisotropy of the charge distribution on the cluster core see Hernandez Sanchez Angew. Chem. Int. Ed. 2018)

Minor point: Ref. 15 is incomplete. 

Author Response

General comment: Electron transfer ability and redox chemistry of molecular forms of electron-rich cluster compounds are at the heart of a fruitful initiative that aim at properly re-assemble these superatoms into metal-organic constructs, especially for chalcogenide species taken off extended solids prior to exploring their fate once made available in polar solvents.  This post-high-temperature solid state chemistry initiative continue to be rewarded by a rich superatomic solid state chemistry  and physics. Yet, just how to tune the redox character thereby the amount of charge transferred remain a recurrent core issue throughout this journey. This is precisely the purpose of this paper. It is timely as the author note, supertonic solids are becoming influential. The paper will attract attention in the field.

Answer on general comment: We greatly appreciate the positive comments of the Referee.

Main remark: the authors stress that the effective charge localized on the TM6 core if key to control the redox potential.  It seems to me there is an opportunity for real significance further advances by exploring how  the electronic parameters provided by the g-factor of the 23 e- species are affected by the effective ionic charge, and its distribution over the 8 inner sites. For an investigation of the anisotropy of the charge distribution on the cluster core see Hernandez Sanchez Angew. Chem. Int. Ed. 2018)

Answer on the main remark: We agree with the Referee that the BVS calculations performed not only for 24 electron cluster compounds, but also for the 23 electron species might advance our knowledge on the oxidation process in cluster compounds. The problem is that the BVS calculations are based on the structural data, which are very rare for the 23 electron clusters. This is the reason why we limited our study to 24 electron clusters.

We thank the Referee for the reference devoted to the anisotropy of the charge distribution in cluster compounds with Co6Se8-core. As far as we understand, the anisotropy described by Hernandez Sanchez et al. is related to different composition of the outer ligands, and it results in the charge asymmetry in the Co6-clusters. In contrast, the Re6-, Mo6- and W6-clusters under study are highly symmetric, and our study is focused on the difference between ionic charges of the inner and outer ligands. However, in future it might be interesting to compare the results of the BVS calculations and those performed by Hernandez Sanchez et al.

Minor point: Ref. 15 is incomplete.

Answer: The reference was completed.